# Effects of ethnicity and geography on the fecal microbiota and dietary habits of Tibeto-Burman hill tribes in Northern Thailand

Tanapon Seetaraso[1,2], Lucsame Gruneck[3], Vasana Jinatham[3,4], Sitanan Kantakat[1], Marie-Lou Albert[3], Phatthanaphong Therdtatha[5], Angkhana Inta[1], Metawee Srikummool[6,7], Jatupol Kampuansai[1]*, Siam Popluechai[3,4]*

1 Department of Biology, Faculty of Science, Chiang Mai University, Chiang Mai, Thailand, 2 Ph.D.'s Degree Program in Biology (International Program), Faculty of Science, Chiang Mai University, Chiang Mai, Thailand, 3 Gut Microbiome Research Group, Mae Fah Luang University, Chiang Rai, Thailand, 4 School of Science, Mae Fah Luang University, Chiang Rai, Thailand, 5 Specialized Research in Microbiome and Metabolome for Health Laboratory, Division of Biotechnology, Faculty of Agro-Industry, Chiang Mai University, Chiang Mai, Thailand, 6 Department of Biochemistry, Faculty of Medical Science, Naresuan University, Phitsanulok, Thailand, 7 Center of Excellence in Medical Biotechnology, Faculty of Medical Science, Naresuan University, Phitsanulok, Thailand

* Jatupol.K@cmu.ac.th (JK); siam@mfu.ac.th (SP)

## Abstract

Numerous studies have shown that both ethnicity and geography influence gut microbiota composition; however, the effects of these parameters remain under-studied in Thailand. We explored the fecal microbiota of 102 individuals, from the Tibeto-Burman-speaking hill-tribe populations (Akha, Lahu, and Lisu) residing in two provinces of Northern Thailand, Chiang Mai, and Chiang Rai, using quantitative PCR polymerase chain reaction. Multivariate statistical analyses, including multiple factor analysis and partial least squares discriminant analysis, were conducted to explore associations between microbiota composition, ethnicity, geographic location, and other host variables (dietary behaviors and participant characteristics). Both ethnicity and geography were linked to gut microbiota composition and dietary patterns, with geography exhibiting a stronger association with variations in microbiota. Ethnicity, however, was primarily related to differences in dietary habits. Notably, gut microbiota profiles were more closely aligned among ethnic groups within the same location than among individuals of the same ethnicity from different regions. The relationship between diet and gut microbiota varied across ethnic and geographic groups, while host factors had a relatively minor impact on microbiota variation. These findings contribute to a broader understanding of variations in gut microbiota among ethnic groups in Northern Thailand and highlight a notable association between geographic location and gut microbiota composition.

**Data availability statement:** All relevant data are within the paper and its Supporting Information files.

**Funding:** This work was supported by the Fundamental Fund 2024, Chiang Mai University (Grant No. FF054/2567 to AI; website: https://www.tsri.or.th), Mae Fah Luang University (Grant No. F24-681G-01-007 to SP; website: https://www.mfu.ac.th), and the Thailand Graduate Institute of Science and Technology, National Science and Technology Development Agency (Grant No. SCA-CO-2565-17158-TH/2565 to TS; website: https://www.nstda.or.th/tgist). The funders had no role in study design, data collection and analysis, decision to publish, or preparation of the manuscript.

**Competing interests:** The authors declare no competing interest.

## Introduction

The human gut microbiota consists of trillions of microorganisms, including bacteria, fungi, archaea, and viruses, that reside in the digestive tract, primarily in the intestines [1,2]. These microorganisms play diverse roles in host health, including digestion, immune homeostasis, colonization resistance against pathogens and metabolism. The gut microbiota is mainly composed of two major phyla: Firmicutes and Bacteroidetes. Firmicutes includes common genera such as *Clostridium*, *Enterococcus*, *Lactobacillus*, and *Ruminococcus*, while Bacteroidetes includes *Bacteroides* and *Prevotella*. Other phyla, such as Actinobacteria, Proteobacteria, Verrucomicrobia, and Euryarchaeota, are present in smaller amounts [3,4]. Variations in gut microbiota are shaped by host and environmental factors, such as genetic background, diet, lifestyle, disease, medications, and gender [5–8]. Many studies suggest that dietary intake is the most influential parameter affecting the gut microbiota. For example, fiber-rich diets tend to promote a beneficial microbial composition, including an increased abundance of *Bifidobacterium* and *Prevotella*, while Western diets tend to promote *Ruminococcus* [9,10].

While diet is recognized as one of the most influential modulators of gut microbiota, the interplay between ethnicity and geographic location further shapes dietary behaviors, which in turn influence gut microbiota composition [11,12]. Ethnicity often dictates cultural and traditional dietary practices, shaping long-term food consumption patterns and preferences [13,14]. Previous studies have shown distinct gut microbiota profiles between different ethnic groups, such as Chinese, Indian, Malay, and Jakun populations in Peninsular Malaysia [15], as well as between multi-ethnic populations in Amsterdam [16]. Geographic location also contributes to variations in gut microbiota by influencing local food availability, dietary habits, and environmental microbial exposure [12,17–20]. A recent comparative study between Han and non-Han populations across seven Chinese provinces found that geography had a greater impact on gut microbiota composition than ethnicity alone [21]. Thus, the interaction between ethnicity, geographic location, and dietary behavior plays a crucial role in determining gut microbiota composition, highlighting the need to study these factors together to better understand their combined effects.

Hill tribes are various ethnic groups that primarily inhabit the high mountainous regions of Northern and Western Thailand. These groups can be classified into three primary linguistic families based on their spoken language: the Sino-Tibetan, Austroasiatic, and Hmong-Mien populations [22]. One of the major branches of the Sino-Tibetan family is the Tibeto-Burman-speaking population, which constitutes most of the hill tribe communities in Thailand, including the Karen, Akha, Lahu, and Lisu ethnic groups [23]. Their original homeland is believed to be Mongolia and the eastern regions of the Tibetan Plateau. Subsequently, they migrated to western China and further to evergreen forests, settling in areas approximately 500 m above sea level in Myanmar and Thailand [24,25]. A previous study on school-aged children from hill tribe backgrounds investigated the association between ethnicity and gut microbiota composition among five ethnic groups: Akha, Chinese, Lahu, Thai, and Thai Yai in

Chiang Rai Province of Northern Thailand. The results indicated that ethnicity, although modest in influence, contributed to gut microbiota composition [26]. However, research investigating the effects of ethnicity and geography on the gut microbiome in Thailand remains limited, with existing studies constrained by small sample sizes, geographically restricted cohorts, and a focus on children, leaving adult populations underrepresented.

To address this gap, the present study is, to the best of our knowledge, the first to examine the effects of ethnicity and geography on the gut microbiota composition of adult Tibeto-Burman-speaking hill-tribe populations in Northern Thailand using quantitative polymerase chain reaction (qPCR) for absolute quantification and comparison of specific gut bacterial taxa. We accounted for ethnic backgrounds (Akha, Lahu, and Lisu) and geographic regions (Chiang Mai and Chiang Rai Provinces) and employed multivariate analyses to explore the associations between multiple variables and variation in gut microbiota. Our findings provide preliminary insights into how ethnic and geographic factors are associated with the gut microbiota in these underrepresented communities, contributing valuable data for future research on the health of populations in Thailand.

## Materials & methods

### Ethical statement

This study was approved by the Ethics Committee of the Faculty of Medicine, Chiang Mai University (STUDY CODE: NONE-2565–08787, Research ID: 8787). Participant rights and confidentiality were rigorously protected throughout the study. All procedures adhered strictly to relevant ethical guidelines and regulations, in accordance with the Declaration of Helsinki. Written informed consent was obtained from all participants prior to interviews and sample collection, either by signature or, for those unable to sign, by fingerprint in the presence of an impartial witness.

### Study population

We collected samples from 102 unrelated individuals residing in high-altitude environments (> 600 m above sea level) from five Tibeto-Burman-speaking villages in Northern Thailand (Table 1). In Chiang Mai Province, the sampled individuals belonged to the Akha and Lahu ethnic groups, while in Chiang Rai Province, participants were from the Akha, Lahu, and Lisu communities. These ethnic groups were selected based on historical and ethnographic evidence from previous

Table 1. Sampling information.

| No. | Code[a] | Ethnic group | Sample size | Location[b] | Latitude[c] | Longitude[c] | Elevation[d] |
|---|---|---|---|---|---|---|---|
| 1 | AkhaCM | Akha | 22 | Ban A-Yae, Phrao, Chiang Mai, Thailand | 19.45 | 99.30 | 861 |
| 2 | AkhaCR | Akha | 23 | Ban A-Bae, Mae Fah Luang, Chiang Rai, Thailand | 20.15 | 99.68 | 1,137 |
| 3 | LahuCM | Lahu | 19 | Ban Khon Muang, Phrao, Chiang Mai, Thailand | 19.46 | 99.26 | 628 |
| 4 | LahuCR | Lahu | 16 | Ban Ja Buu Si, Mae Fah Luang, Chiang Rai, Thailand | 20.12 | 99.65 | 907 |
| 5 | LisuCR | Lisu | 22 | Ban That, Mae Fah Luang, Chiang Rai, Thailand | 20.12 | 99.68 | 934 |

[a]AkhaCM (Akha ethnic group residing in Chiang Mai province); AkhaCR (Akha ethnic group residing in Chiang Rai province); LahuCM (Lahu ethnic group residing in Chiang Mai province); LahuCR (Lahu ethnic group residing in Chiang Rai province); LisuCR (Lisu ethnic group residing in Chiang Rai province).

[b]Locations correspond to villages where samples were collected.

[c]Latitude and longitude are given in decimal degrees.

[d]Elevation is meters above sea level (ASL).

reports that supported their long-term settlement and cultural distinctiveness in the region [27–29]. Sampling and surveys were conducted from November 2022 to March 2023. To ensure consistency in participant backgrounds, we applied selection criteria established in a previous genetic study [30]. Only adult volunteers aged over 20 years with no known ancestry from other ethnic groups within the past three generations were included. Personal data, including endonyms, self-reported unrelated lineages, languages, and migration histories, were collected through structured oral interviews. Additionally, all participants reported no use of antibiotics or anthelmintics for at least three months and were not experiencing diarrhea at the time of sampling.

## Characteristic factors and dietary behavior questionnaires

Anthropometric and physiological measurements including gender, age, weight, height, waist circumference, hip circumference, systolic blood pressure (SBP), and diastolic blood pressure (DBP) were recorded at the time of sample collection using standardized instruments. Body mass index (BMI) was calculated as weight (kg) divided by height squared ($m^2$), while the waist-to-hip ratio (WHR) was determined by dividing waist circumference (cm) by hip circumference (cm). Dietary behavior data were collected using a Food Frequency Questionnaire, comprising 20 items that represented both non-processed (ten items) and processed (ten items) foods. This frequency-based questionnaire reflected general daily consumption behaviors across ethnicities. Consumption frequency was categorized into six levels: daily (6), 5–6 times a week (5), 3–4 times a week (4), 1–2 times a week (3), less than once a week (2), and never (1). Differences in mean ranks among groups were assessed using the Kruskal-Wallis rank sum test, followed by Dunn's post-hoc test for multiple comparisons, with $p$–values adjusted using the Benjamini–Hochberg (BH) method.

## Sample collection, DNA extraction, and qPCR

After fasting for at least 8 h, stool samples were collected from the volunteers and immediately stored in sterilized containers at −80°C. Microbiota DNA was extracted from stool samples using the QIAmp Fast DNA Stool Mini Kit (Qiagen, Hilden, Germany) following the manufacturer's protocol. DNA yield and purity were determined using the Take 3 Micro-Volume Plate (Biotek, Winooski, VT, USA). Absolute quantification of bacteria was performed using qPCR on Real-Time Thermal Cyclers CFX96 Touch™ (Bio-Rad, Singapore). The primers targeting microbiota 16s rRNA genes used in this study are summarized in S1 Table in S2 Text. Each reaction contained template DNA, forward and reverse primers, 1X SYBR Green (2X SensiFAST™ SYBR No-ROX mix, BIOLINE, UK), and nuclease-free water. Assay conditions and microbiota copy number calculations were conducted according to a previously described protocol [26,31]. The average estimates of microbiota abundance, obtained by converting cycle threshold values, were expressed as the logarithmic copy number per gram of wet-weight feces.

## Direct statistical analysis

In this study, statistical analyses were conducted at two primary levels: ethnicity (Akha, Lahu, and Lisu) and geographic location [Chiang Mai (CM) and Chiang Rai (CR) Provinces]. Ethnicity-based sublevel comparisons were performed within Chiang Mai province [AkhaCM (Akha ethnic group residing in Chiang Mai Province) vs. LahuCM (Lahu ethnic group residing in Chiang Mai Province)] and within Chiang Rai Province [AkhaCR (Akha ethnic group residing in Chiang Rai Province) vs. LahuCR (Lahu ethnic group residing in Chiang Rai Province) vs. LisuCR (Lisu ethnic group residing in Chiang Rai Province)]. Geographic sublevel comparisons were conducted within the Akha ethnic group (AkhaCM vs. AkhaCR) and within the Lahu ethnic group (LahuCM vs. LahuCR). The raw data, including group categories of ethnicities and geographic locations, participant characteristic factors, dietary behavior data, and microbial abundance (log10 copy number per gram of wet weight feces), are provided in S1 File in S3 Text. All analyses were performed using the R software version 4.4.3 [32]. Normality and homogeneity of variance were assessed using the Shapiro–Wilk test and Levene's test, respectively (stats package version 4.4.3) [32]. Statistical methods were selected based on data distribution. Differences in gender proportion between or across groups were determined using Chi-square and Fisher's exact

test (rcompanion package version 2.5.0) [33]. Differences in the abundance of gut microbiota between or across groups were determined using unpaired two-tailed Student's t-test, Welch's t-test, the Mann–Whitney test, one-way analysis of variance (ANOVA), Welch's ANOVA, and the Kruskal–Wallis test, followed by multiple comparisons using Tukey's post-hoc test, the Games-Howell test (PMCMRplus package version 1.9.12) [34], and Dunn's test with BH $p$-value correction (hereafter referred to as $q$-value). Relationships between the absolute abundances of gut microbiota and dietary habits were analyzed using Spearman's rank correlation coefficient.

## Multivariate statistical analysis

Both unsupervised (multiple factor analysis, MFA) and supervised (partial least squares-discriminant analysis, PLS-DA) statistical approaches were conducted to examine individual profiles related to ethnicities, geographic locations, dietary behaviors, and participant characteristics, and to evaluate their influence on gut microbiota abundance. Raw qPCR data and all metadata used in the multivariate analyses are provided in S2–S4 Files in S3 Text. The microbial abundances were visualized as a heatmap using ComplexHeatmap (version 2.22.0), with hierarchical clustering based on a Euclidean distance matrix [35,36]. PLS-DA was conducted using the mixOmics package version 6.30.0 [37] to identify microbial taxa most relevant in distinguishing between groups. Variable importance in projection (VIP) scores were used to assess the contribution of each feature to classification, with VIP scores > 1 indicating important features and VIP scores < 1 indicating less important ones. Receiver operating characteristic (ROC) curves and the area under the curve (AUC) were calculated to assess the validity of the supervised classification results. The classification accuracy of PLS-DA models was interpreted as follows: no discrimination (AUC = 0.5), low discrimination (AUC = 0.6–0.7), acceptable discrimination (AUC = 0.7–0.8), excellent discrimination (AUC = 0.8–0.9), and outstanding discrimination (AUC > 0.9) [38,39]. MFA was conducted using the FactoMineR package version 2.11 [40] to assess the relationships between host variables (dietary behaviors and participant characteristics) and gut microbiota across different ethnicities and geographic locations. Associations were assessed between gut microbiota (24 variables) and dietary behaviors (19 variables, excluding rice, as it was consumed by all individuals), as well as between gut microbiota, dietary habits, and participants characteristics (six variables). All variables were included as active components in the MFA models to ensure that their full contributions were considered. The Factoextra package version 1.0.7 [41] was used to visualize variable contributions and correlations in multidimensional space. Permutational multivariate analysis of variance (PERMANOVA) analysis was conducted using the *adonis2* function (vegan package version 2.6–10) [42] to determine whether overall gut microbiota profile and dietary behaviors differed significantly by ethnicity and geographic locations, adjusting for confounders (such as age, gender, BMI, DBP, SBP) identified as contributing factors in MFA. To ensure comparability, raw dietary data, age, BMI, DBP, and SBP were standardized using z-score transformation prior to analysis. Dissimilarities were computed using a Euclidean distance matrix. Permutational analysis of multivariate dispersion (PERMDISP) was performed using the *betadisper* function with 999 permutations based on a Euclidean distance matrix, as implemented in the *vegan* package (version 2.6–10) [42], to assess group dispersion. Principal coordinates analysis was conducted based on the same distance matrix used for PERMANOVA. Ordination scores were extracted using the *vegan:scores* function and visualized in *ggplot2* (package version 3.5.1) [43] to depict the spread of samples around their group centroids. Collinearity analysis was performed using Variance inflation factors (VIFs) and Generalized variance inflation factors (GVIFs) in R (car package version 3.1–3) [44] to assess potential multicollinearity among predictor variables. Low collinearity, including for multi-categorical variables, was detected, as all VIF and GVIF ($GVIF^{1/(2*Df)}$) values were within a low range (below 5) [45].

## Results

### Participant characteristics

We evaluated differences in age, BMI, WHR, DBP, and SBP among ethnic groups in Chiang Mai and Chiang Rai Provinces. No significant differences were observed in these parameters across ethnic groups within each province. A

significant difference in age was only observed between the AkhaCM and AkhaCR groups ($p = 0.036$), while no significant differences in age, BMI, WHR, DBP, or SBP were found within the Lahu group residing in different locations. (Table 2).

**Analysis of gut microbiota, dietary habits, and participant characteristics across ethnicities**

**Differences in the abundance of gut microbiota associated with ethnicity.** The abundance of gut microbiota across different ethnic groups residing in Chiang Mai and Chiang Rai Provinces was compared. The results indicated a significant variation in microbial abundance among ethnic groups within each geographic location (S2 Table in S2 Text). In Chiang Mai province, a significant difference was detected in Actinobacteria, whose abundance was significantly higher in AkhaCM than in LahuCM ($q = 0.044$; Fig 1). PLS-DA analysis, employed to identify key discriminatory microbial taxa, revealed that Actinobacteria (VIP = 2.52) and *Bacteroides fragilis* (VIP = 2.21) were the most discriminative taxa in the Akha ethnic group, whereas *Enterococcus* spp. (VIP = 1.16) and *Staphylococcus* spp. (VIP = 1.32) were enriched in individuals of the Lahu ethnic group in Chiang Mai Province (Component 1: 27% explained variance, AUC = 0.82, $p < 0.001$; S1A Fig in S1 Text). In LahuCM, *Coprococcus* spp. (VIP = 1.05) was the most discriminative bacterium in (Component 2: 29% explained variance, AUC = 0.88, $p < 0.001$; S1B Fig in S1 Text).

In Chiang Rai Province, significant differences were observed in the abundance of Firmicutes ($q = 0.038$), *Gammaproteobacteria* ($q = 0.033$), *Ruminococcus* ($q = 0.01$), *Actinomyces* ($q < 0.001$), and *Christensenella minuta* ($q = 0.008$), which were more abundant in AkhaCR than in LisuCR (S3 Table in S2 Text). Conversely, *Actinomyces* ($q = 0.002$) and *Lactobacillus* ($q < 0.001$) were more abundant in LahuCR than in LisuCR. The results aligned with the PLS-DA, clearly distinguishing the enrichment of *Actinomyces* (VIP = 2.01) in AkhaCR from the abundance of that genus in LahuCR and LisuCR (Component 1, 36% explained variance, AUC = 0.68, $p = 0.023$) (S1C Fig in S1 Text). Both direct comparisons and supervised analyses consistently highlighted the enrichment of *Actinomyces*, a genus within the Actinobacteria phylum, as being enriched the most in AkhaCR, whereas this particular gut bacterium was least prevalent in LisuCR.

**Association between gut microbiota and dietary habits.** We assessed the frequency-based questionnaire of dietary habits, which reflected general daily consumption behaviors across ethnicities. Individuals from different ethnic backgrounds exhibited variations in dietary consumption. In Chiang Mai Province, the consumption of fresh meat ($q = 0.002$), vegetables ($q = 0.002$), and milk ($q = 0.044$) was significantly higher in AkhaCM than in LahuCM (Fig 2 and

**Table 2. Participants characteristics of the study population.**

| Factor | Total | AkhaCM | AkhaCR | LahuCM | LahuCR | LisuCR |
|---|---|---|---|---|---|---|
| | (n = 102) | (n = 22) | (n = 23) | (n = 19) | (n = 16) | (n = 22) |
| **Gender** | | | | | | |
| Male | 39 (38.24%) | 7 (31.82%) | 9 (39.13%) | 5 (39.13%) | 8 (50.00%) | 10 (45.5%) |
| Female | 63 (61.76%) | 15 (68.18%) | 14 (60.87%) | 14 (60.87%) | 8 (50.00%) | 12 (54.5%) |
| **Age (years)** | 48.15 ± 13.14 | 52.95 ± 14.24[a] | 44.74 ± 11.09[a] | 51.79 ± 11.93 | 44.94 ± 14.78 | 46.09 ± 12.4 |
| **BMI (kg/m²)** | 25.60 ± 4.38 | 23.73 ± 3.49 | 26.01 ± 4.24 | 26.01 ± 4.00 | 24.79 ± 4.81 | 27.29 ± 4.68 |
| **Waist-to-hip ratio (WHR)** | 0.86 ± 0.05 | 0.84 ± 0.05 | 0.87 ± 0.06 | 0.86 ± 0.05 | 0.86 ± 0.05 | 0.87 ± 0.06 |
| **Systolic BP (mmHg)** | 126.54 ± 16.97 | 125.95 ± 17.47 | 127.13 ± 18.45 | 125.42 ± 12.29 | 124.81 ± 16.81 | 128.73 ± 19.9 |
| **Diastolic BP (mmHg)** | 80.18 ± 9.89 | 81.32 ± 11.18 | 82.74 ± 9.44 | 79.05 ± 8.4 | 80.25 ± 7.13 | 77.27 ± 11.73 |

[a]AkhaCR < AkhaCM. The age of AkhaCR was significantly lower than that of AkhaCM ($p = 0.036$). Statistical significance was assessed using the student's t-test. CM = Chiang Mai; CR = Chiang Rai; BP = Blood pressure.

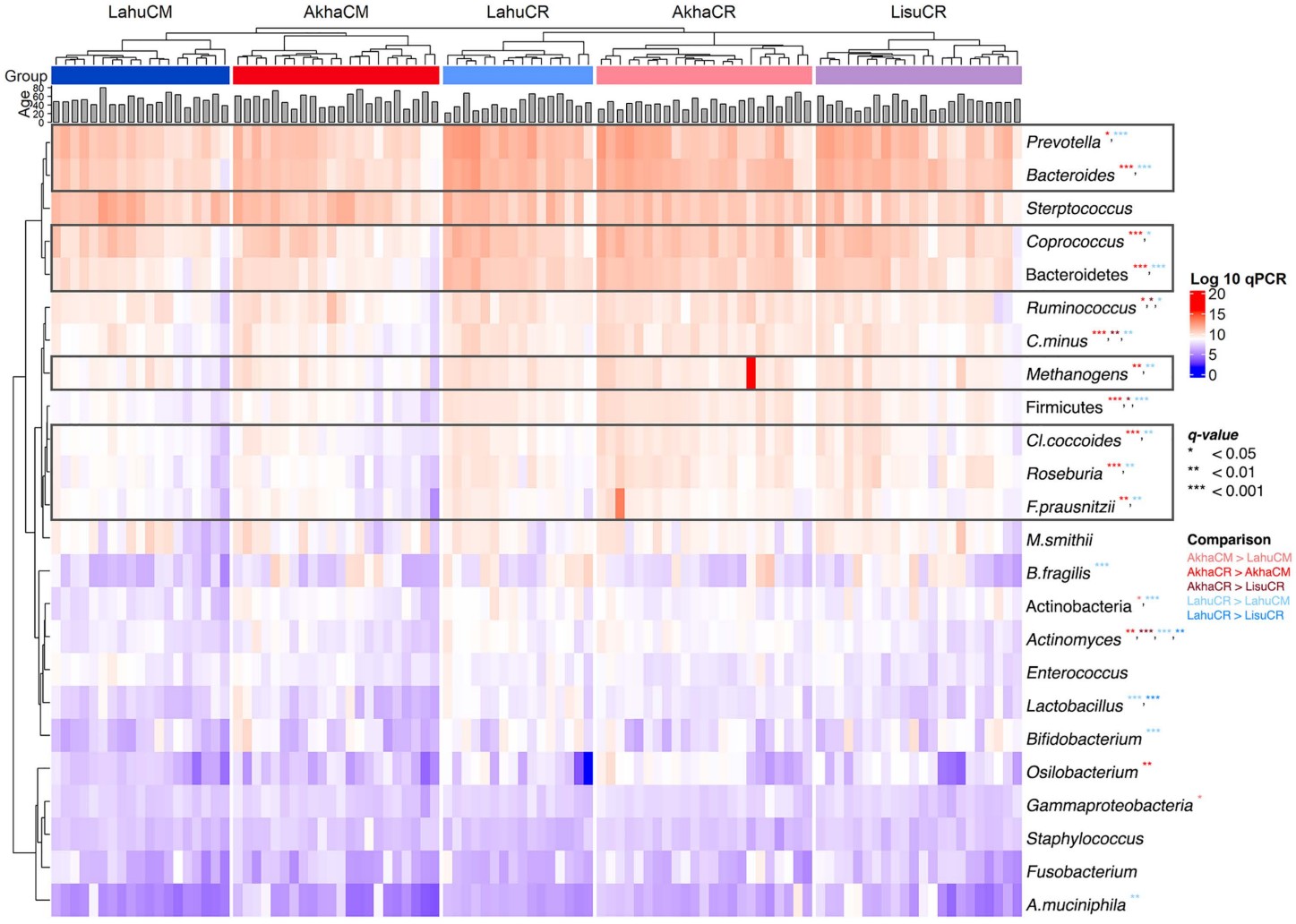

**Fig 1. Hierarchical clustering of gut microbiota based on a Euclidean distance matrix across ethnicities and geographic locations, with age annotations.** The heat map displays normalized bacterial abundances (log10 qPCR 16S rRNA copy number per gram of feces, wet weight). Grey bar charts represent the age of participants. The abundance of microbial taxa is illustrated using a blue-to-red color gradient. Asterisks (*) indicate taxa with significant differences between groups after Benjamini–Hochberg correction (*$q<0.05$; **$q<0.01$; ***$q<0.0001$). Grey rectangles highlight specific taxa that were more abundant in CR compared to CM and were associated exclusively with geographic location, controlling for ethnicity. AkhaCM = Akha ethnic group residing in Chiang Mai province; AkhaCR = Akha ethnic group residing in Chiang Rai province; LahuCM = Lahu ethnic group residing in Chiang Mai province; LahuCR = Lahu ethnic group residing in Chiang Rai province; LisuCR = Lisu ethnic group residing in Chiang Rai province. CM = Chiang Mai; CR = Chiang Rai.

S4 Table in S2 Text). In Chiang Rai Province, AkhaCR consumed more non-processed foods ($q=0.02$) and less bread ($q=0.017$) than LahuCR. Fresh meat consumption was significantly higher by AkhaCR than by LahuCR ($q=0.001$) and by LisuCR than by LahuCR ($q<0.001$). Similarly, egg consumption was greater by AkhaCR than by LahuCR ($q=0.009$) and by LisuCR than by LahuCR ($q=0.032$). Grain consumption was higher in LisuCR compared to AkhaCR ($q=0.021$), while sticky rice was consumed the least by AkhaCR ($q<0.001$; Fig 2 and S5 Table in S2 Text). Correlations between dietary habits and gut microbiota further revealed moderate negative correlations between egg consumption and *Streptococcus* spp. ($r=-0.58$, $q=0.047$) and between vegetable consumption and *Fusobacterium* spp. ($r=-0.59$, $q=0.040$) in AkhaCM. Fresh meat consumption exhibited a strong negative correlation with *C. minuta* among LahuCM individuals ($r=-0.65$,

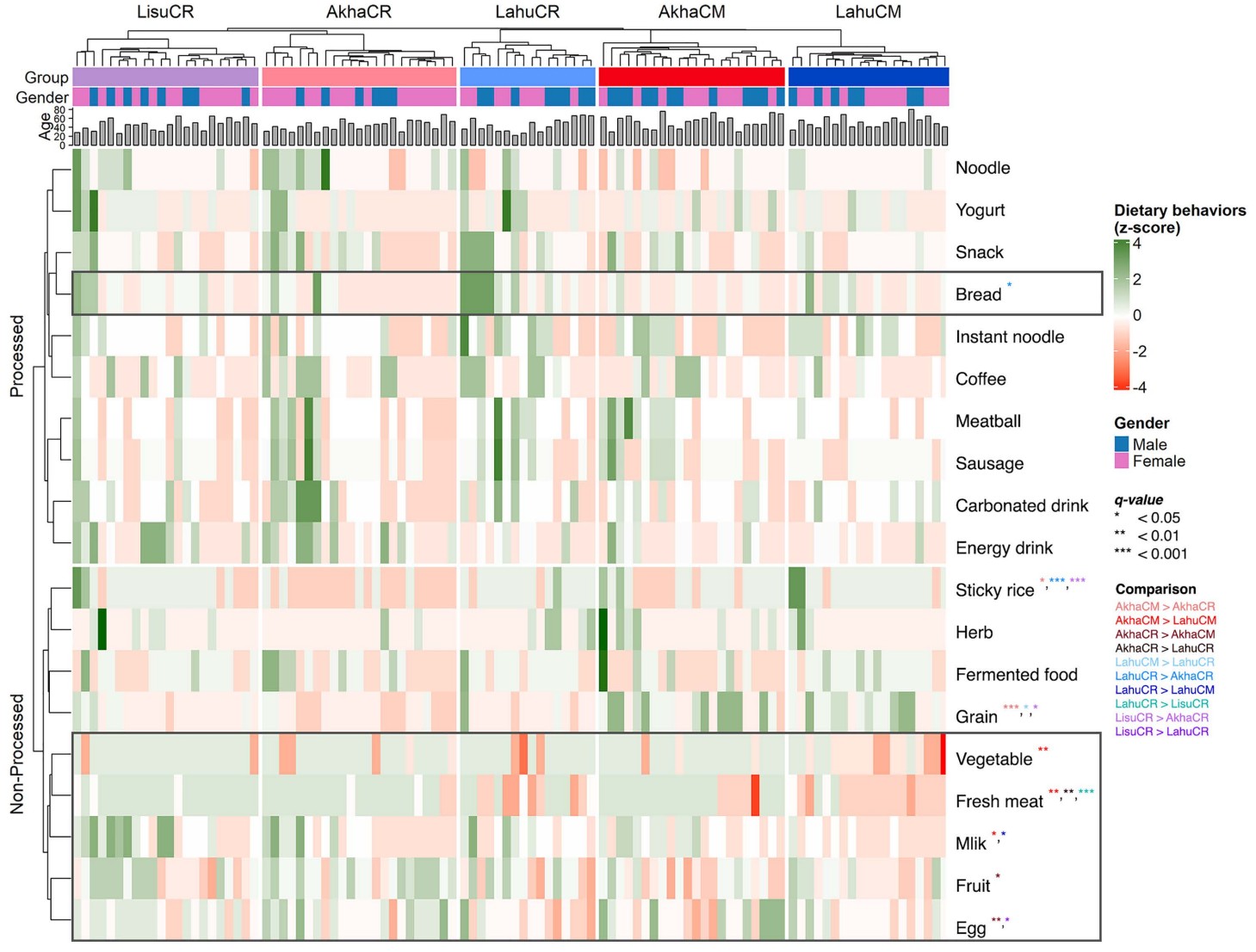

**Fig 2. Hierarchical clustering of dietary behaviors based on a Euclidean distance matrix across ethnicities and geographic locations, annotated with participant age and gender.** Row clustering differentiates non-processed and processed food groups. The heatmap displays standardized (z-score) values for 19 food items (excluding rice, as it was consumed by all individuals), visualized using a red-to-green color gradient. Grey bar charts indicate participant ages, while gender is denoted by blue (male) and pink (female) color keys. Asterisks (*) indicate dietary habits with significant differences between groups after Benjamini–Hochberg correction (*$q < 0.05$; **$q < 0.01$; ***$q < 0.0001$). Grey rectangles highlight specific dietary habits with significant differences associated exclusively with ethnicity. CM = Chiang Mai; CR = Chiang Rai.

$q = 0.048$; S6 Table in S2 Text). In Chiang Rai Province, energy drink and fresh meat consumption showed a strong positive correlation with *Methanobrevibacter smithii* ($r = 0.79$, $q < 0.001$) and *Faecalibacterium prausnitzii* ($r = 0.62$, $q = 0.040$) in AkhaCR. Fruit consumption showed the strongest negative correlation with *Fusobacterium* spp. ($r = -0.75$, $q = 0.027$) among LahuCR individuals. Sausage consumption showed a strong positive correlation with *B. fragilis* ($r = 0.63$, $q = 0.039$), whereas instant noodle consumption displayed a strong negative correlation with *Enterococcus* spp. ($r = -0.62$, $q = 0.041$) in LisuCR (S7 Table in S2 Text). Notably, the Akha ethnic group exhibited a consistently higher fresh meat consumption than the Lahu group, regardless of geographic location.

Further integration of dietary habits and gut microbiota profiles using MFA showed no strong associations between gut microbiota and dietary habits among ethnic groups in Chiang Mai Province (S2 Fig in S1 Text). Instead, ethnicity accounted for a greater proportion of dietary variation at this location, particularly along Dim 2 (explained variance: 11.56%, contribution: 38.02%). AkhaCM exhibited a higher consumption trend for fresh meat, milk, and vegetables than LahuCM. A similar pattern was observed in Chiang Rai Province, where ethnicity contributed substantially to dietary variations in Dim 2 (40.94% contribution), whereas the influence of ethnicity on variations in gut microbiota was relatively low in Dim 1 (6% contribution; S3 Fig in S1 Text). Consumption of energy drinks ($r = 0.56$, $p < 0.001$) was less prevalent in LahuCR (coordinate = 1.73, $p < 0.001$) than in AkhaCR (coordinate = −1.25, $p < 0.001$) in Dim 2 (9.75%), while bread consumption ($r = 0.55$, $p < 0.001$) showed the opposite trend. In Dim 3 (7.88%), LisuCR (coordinate = 1.02, $p < 0.001$) exhibited a distinct association pattern in dietary habits and gut microbiota compared to AkhaCR (coordinate = −0.53, $p = 0.004$) and LahuCR (coordinate = −0.53, $p < 0.001$), characterized by lower coffee consumption ($r = −0.46$, $p < 0.001$) and increased abundance of *M. smithii* ($r = 0.46$, $p < 0.001$). These two variables were nevertheless moderately correlated to Dim 3. These results demonstrate that ethnicity was predominantly associated with variations in dietary patterns, while its influence on gut microbiota was less pronounced when both variables (dietary habits and gut microbiota) were considered together.

**Associations between gut microbiota, dietary habits, and participants characteristics.** Participant characteristics were also taken into account to evaluate their influence, in conjunction with dietary habits, on gut microbiota composition. Integration of these variables using MFA revealed no association among these factors between the two ethnic groups within Chiang Mai Province (Dim 1: 20.15%; S4 Fig in S1 Text). The contribution of ethnicity and gender to variations in dietary habits was stronger than that of gut microbiota, which was captured by Dim 2 (10.17%) and Dim 3 (9.34%), respectively. LahuCM (coordinate = 0.99, $p < 0.001$) displayed a contrasting profile to AkhaCM (coordinate = −0.99, $p < 0.001$) in Dim 2, where BMI ($r = 0.64$, $p < 0.001$) and instant noodle consumption ($r = 0.53$, $p < 0.001$) were positively associated with LahuCM. These two variables also showed a positive trend among females (coordinate = 0.75, $p = 0.001$). In contrast, milk ($r = −0.53$, $p = 0.0037$) and fresh meat consumption ($r = −0.51$, $p < 0.001$) were higher among AkhaCM and males (coordinate = −0.75, $p = 0.001$). However, gender contributed more to Dim 3, where males and females exhibited contrasting dietary habits, particularly in the consumption of coffee ($r = −0.50$, $p < 0.001$), grain ($r = −0.48$, $p = 0.001$), and energy drinks ($r = 0.46$, $p = 0.002$). The first two dietary components were consumed more by females (coordinate = −1.02, $p < 0.001$), whereas energy drinks were consumed more by males (coordinate = 1.02, $p < 0.001$). While MFA emphasized a stronger contribution of ethnicity than gender, PERMANOVA revealed that dietary habits also differed significantly between the ethnic groups ($F_{by\ margin} = 2.19$, $p = 0.005$) after adjusting for BMI and gender. However, significant group dispersion was observed, particularly within the AkhaCM group, where dietary patterns exhibited greater variability ($F_{PERMDISP} = 9.29$, $p = 0.005$; S5 Fig in S1 Text). This indicated that unequal within-group dispersions may have partially influenced the significant PERMANOVA results. Additionally, a significant interaction between ethnicity and gender ($F_{by\ term} = 1.53$, $p = 0.006$) as well as between ethnicity and BMI ($F_{by\ term} = 1.49$, $p = 0.02$) was detected; however, the effect sizes of these interactions on microbial abundance were relatively small across ethnic groups. Moreover, no significant differences in the overall gut microbiota profile were observed within Chiang Mai Province.

Furthermore, the variations patterns in gut microbiota and dietary habits with ethnicity in Dim 1 (17.04%) and Dim 2 (8.37%) remained consistent for individuals within the Chiang Rai Province (S6 Fig in S1 Text). The influence of gender was independently captured by Dim 3 (7.75%), where males (coordinate = 1.02, $p < 0.001$) consumed more coffee ($r = 0.51$, $p < 0.001$) and less yogurt ($r = −0.49$, $p < 0.001$) than females (coordinate = −1.02, $p < 0.001$). In Dim 4 (7.21%), which explained the association between dietary habits and gut microbiota, grain consumption ($r = 0.46$, $p < 0.001$) showed an inverse relationship with *Gammaproteobacteria* ($r = −0.46$, $p < 0.001$). While grain consumption in LisuCR was higher, the abundance of this bacterial class tended to decrease (coordinate = 1.04, $p < 0.001$). This relationship was reversed in AkhaCR (coordinate = −0.82, $p < 0.001$). Additionally, PERMANOVA demonstrated that ethnicity exerted a stronger influence

on variations in overall dietary habits ($F_{by\ margin}$ = 2.35, $p$ = 0.001) than on gut microbiota ($F_{by\ margin}$ = 1.75, $p$ = 0.046) after adjusting for participant characteristics (age, BMI, gender, and DBP), with no significant dispersion. Pairwise comparisons revealed that dietary habits of LahuCR differed significantly from those of AkhaCR ($F_{pairwise}$ = 3.11, $q$ = 0.01) and LisuCR ($F_{pairwise}$ = 2.33, $q$ = 0.02), whereas AkhaCR displayed distinct gut microbiota profiles compared to LahuCR ($F_{pairwise}$ = 2.26, $q$ = 0.046) and LisuCR ($F_{pairwise}$ = 2.41, $q$ = 0.047). While age and BMI were independently (both by terms and by margin) associated with dietary habits, these confounders did not exert an independent effect on gut microbiota composition (by margin). After adjusting for ethnicity, gender, BMI, and DBP, age exhibited a greater independent effect on overall dietary patterns among individuals from Chiang Rai Province ($F_{by\ margin}$ = 5.46, $p$ = 0.001). Specifically, age was negatively associated with the consumption of four out of ten processed food items (carbonated drinks, bread, instant noodles, and yogurt) indicating a declining trend in the consumption of processed food among older individuals in this province, irrespective of ethnicity. When considering interaction effects, age exhibited a moderate effect on dietary habits ($F_{interaction}$ = 2.43, $p$ = 0.001), whereas BMI ($F_{interaction}$ = 1.75, $p$ = 0.003), gender ($F_{interaction}$ = 1.69, $p$ = 0.005), and DBP ($F_{interaction}$ = 1.53, $p$ = 0.008) had relatively small effects. In addition, age ($F_{interaction}$ = 1.70, $p$ = 0.01) showed a small interaction effect with ethnicity in relation to microbial abundance.

Considering all factors together, ethnicity was associated with variations in the overall dietary patterns within Chiang Mai Province, and with both dietary patterns and gut microbiota composition within Chiang Rai Province. These results indicated a possible link between ethnicity and variations in dietary habits.

## Analysis of gut microbiota, dietary habits, and participants characteristics across geographic locations

**Differences in the abundance of gut microbiota associated with geographic location.** The gut microbiota profiles were compared among ethnic groups residing in different geographic locations. Significant differences were observed in 13 taxa, including two phyla, seven genera, three species, and one archaeon, all of which were significantly more abundant in AkhaCR (Fig 1 and S8 Table in S2 Text). PLS-DA highlighted the enrichment of these taxa (such as *Roseburia*, Firmicutes, and Bacteoidetes; VIP = 1.65, 1.64 and 1.61, respectively) in AkhaCR, distinctly separating them from those in AkhaCM (Component 1: 46% explained variance, AUC = 0.91, $p$ < 0.001; Fig 3A). In the Lahu ethnic group, 17 taxa–including three phyla, eight genera, six species, and one archaeon–were significantly more abundant in LahuCR compared to LahuCM (S9 Table in S2 Text). PLS-DA also illustrated distinct differences in microbiota enrichment between LahuCR and LahuCM, with Firmicutes (VIP = 1.58), Bacteroidetes (VIP = 1.40), and *Bacteroides* spp. (VIP = 1.35) being the top three taxa distinguishing LahuCR from LahuCM (AUC = 0.96, $p$ < 0.001; Fig 3B). Notably, both AkhaCR and LahuCR shared a similar trend of higher gut bacterial abundance among the top five taxa–Bacteroidetes, *Streptococcus*, *Coprococcus*, *Prevotella*, and *Bacteroides*–compared to the ethnic groups in Chiang Mai Province. The combined results, derived from both direct comparisons of gut microbiota abundances and PLS-DA, demonstrated that gut microbiota composition may vary in relation to geographic location.

**Association between gut microbiota and dietary habits.** Dietary habits significantly differed between the ethnic groups based on geographic location. In the Akha group, AkhaCM consumed more grain ($q$ < 0.001) and sticky rice ($q$ = 0.016), whereas fruit consumption was significantly higher in AkhaCR ($q$ = 0.016; Fig 2 and S10 Table in S2 Text). Similarly, within the Lahu ethnic group, grain consumption was also significantly higher in LahuCM ($q$ = 0.012), while processed food ($q$ = 0.012) and milk consumption ($q$ = 0.046) were more prevalent in LahuCR (S11 Table in S2 Text).

MFA, integrating dietary habits and gut microbiota revealed significant variations between geographic locations. A contrasting relationship between gut microbiota profiles and grain consumption contributed to individual variation in Dim 1 (28.08%; S2C data in S2 File in S3 Text). The abundance of the top five gut microbiota taxa including Firmicutes, Bacteroidetes, *Cl. coccoides*, *Roseburia*, and *Bacteroides*, showed an increasing trend in AkhaCR (coordinate = 1.56, $p$ < 0.001), while grain consumption was lower ($r$ = −0.53, $p$ < 0.001). In contrast, AkhaCM exhibited the opposite pattern (coordinate = −1.56, $p$ < 0.001). In Dim 2 (9.94%), egg consumption ($r$ = −0.52, $p$ < 0.001) was negatively associated with Actinobacteria

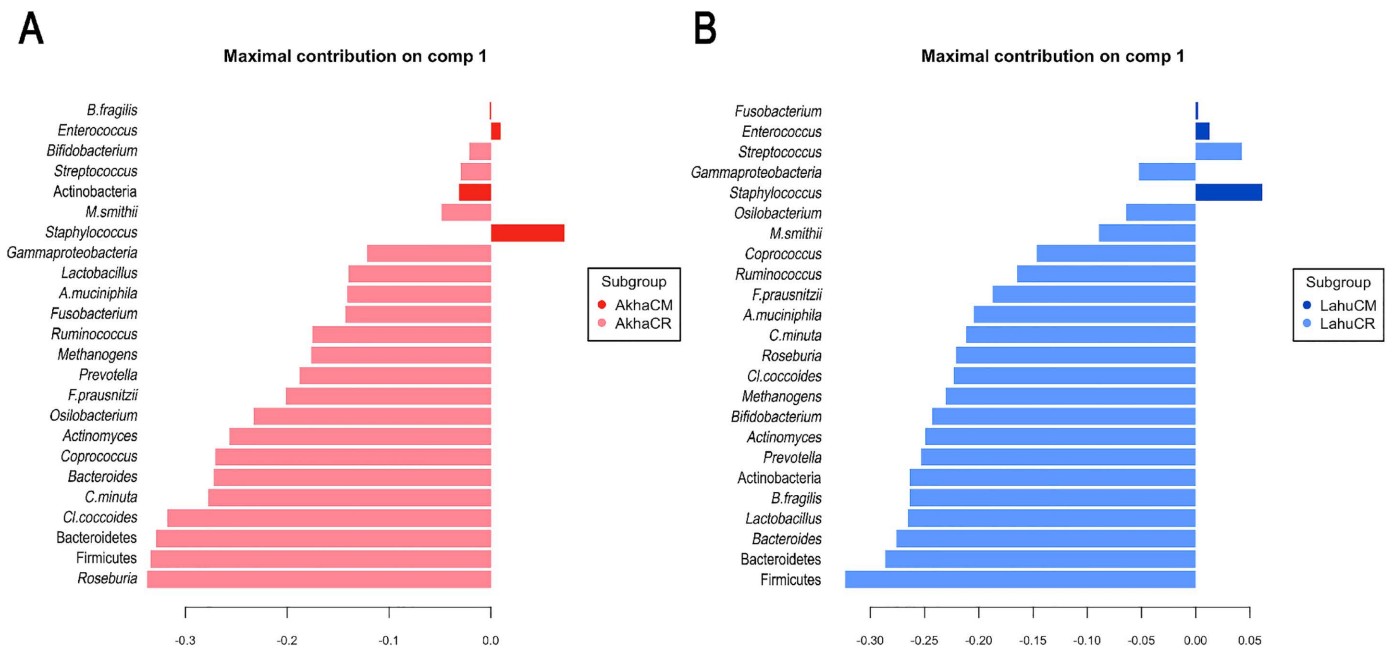

**Fig 3. Partial least squares discriminant analysis (PLS-DA) of gut microbiota between geographic locations.** (A) Discriminant analysis identifying key features (microbial taxa) associated with the Akha ethnic group, where the median values are maximized along Component 1. (B) Discriminant analysis identifying key features (microbial taxa) associated with the Lahu ethnic group, where the median values are maximized along Component 1. Bar lengths represent the loading weights, reflecting the contribution of each taxon (ranked from bottom to top). Red = AkhaCM; Light red = AkhaCR; Blue = LahuCM; Light blue = LahuCR; CM = Chiang Mai; CR = Chiang Rai.

abundance ($r = 0.51$, $p < 0.001$). A lower egg consumption with a higher abundance of Actinobacteria was observed in AkhaCM (coordinate = 0.73, $p < 0.001$) compared to AkhaCR (coordinate = −0.73, $p < 0.001$; S7 Fig in S1 Text). A similar trend in microbial abundance was detected in the Lahu group. Firmicutes, Bacteroidetes, *Bacteroides*, *Cl. coccoides*, and *Methanogens* were more abundant in LahuCR (coordinate = 1.62, $p < 0.001$) than in LahuCM (coordinate = −1.62, $p < 0.001$), as shown in Dim 1 (29.92%; S8 Fig in S1 Text and S2D data in S2 File in S3 Text). In Dim 2 (11.40%), coffee consumption ($r = -0.52$, $p = 0.0011$) and snack consumption ($r = -0.51$, $p = 0.0015$) were negatively correlated with the abundances of *Gammaproteobacteria* ($r = 0.55$, $p < 0.001$) and *Osilobacterium* spp. ($r = 0.54$, $p < 0.001$) in LahuCM (coordinate = 0.63, $p = 0.003$). The inverse trend was observed in LahuCR (coordinate = −0.63, $p = 0.003$). Variation in Dim 2 was more driven by the relationship between coffee consumption and *Gammaproteobacteria*, with a clear contrast between the LahuCM and LahuCR profiles, highlighting association between gut microbiota and dietary habits. In summary, both AkhaCM and LahuCM shared a consistent profile in the prevalence of grain consumption and microbial abundance compared to their counterpart in Chiang Mai Province.

**Associations between gut microbiota, dietary habits, and participants characteristics.** When participant characteristics were integrated into MFA of dietary habits and gut microbiota, geographic location remained the primary determinant of variations in microbial abundance, as reflected in Dim 1 for both the Akha and Lahu ethnic groups. In the Akha group, the association between gut microbiota and grain consumption, along with individual variations between AkhaCR and AkhaCM, remained consistent in Dim 1. In Dim 2 (9.49%), non-processed food was a major contributor to the observed variation, with gender exerting a greater influence than geographic location. Egg consumption was lower ($r = -0.56$, $p < 0.001$), while grain consumption was higher among females ($r = 0.52$, $p < 0.001$; coordinate = 0.85, $p < 0.001$). Males displayed the opposite trend (coordinate = −0.85, $p < 0.001$). Variations in specific taxa, such as *Bifidobacterium*

spp. ($r=0.48$, $p<0.001$) and Actinobacteria ($r=0.46$, $p=0.0013$), were also captured in Dim 2, with a higher abundance trend observed in females. In Dim 3 (8.30%), gender was the primary driver of variations in energy drink consumption ($r=0.52$, $p<0.001$), with males (coordinate = 0.85, $p<0.001$) consuming more than females (coordinate = −0.85, $p<0.001$; Fig 4A-4C). Furthermore, grain consumption appeared to have a more stable and independent influence across different MFA models. These results highlighted the influence of geographic location on both gut microbiota composition and dietary preferences, while also pointing toward a link between gender and dietary habits. Moreover, PERMANOVA confirmed the independent effect of geographic location on microbial abundance ($F_{by\ margin}=4.93$, $p=0.001$) and dietary habits ($F_{by\ margin}=2.92$, $p=0.001$) after adjusting for all other factors (age, gender, BMI, SBP, and DBP), with a stronger effect on gut microbiota and no significant group dispersion. An interaction effect between geographic location and age on dietary habit was also detected ($F_{interaction}=2.31$, $p=0.001$). Age, which significantly differed between AkhaCM and AkhaCR, exhibited an independent influence on dietary habits both by term ($F_{by\ term}=3.01$, $p=0.001$) and by margin ($F_{by\ margin}=2.78$, $p=0.002$). Notably, bread consumption ($r=−0.65$, $q=0.02$) and the relative abundance of *Roseburia* ($r=−0.66$, $q=0.007$) exhibited a negative correlation with age among AkhaCR individuals, indicating a declining trend in both variables among older participants. Moreover, the interaction effect between geographic location and gender ($F_{interaction}=1.62$, $p=0.01$) demonstrated that gender-based dietary patterns varied by region, as evidenced in the MFA results.

The individual variation patterns in gut microbiota between LahuCR and LahuCM remained consistent in Dim 1 (24.60%; Figs 4D-4F). In Dim 2, the variation explained by gender was stronger than that explained by geographic location, in which females (coordinate = 0.96, $p<0.001$) consumed more yogurt ($r=0.55$, $p<0.001$) than males (coordinate = −0.96, $p<0.001$). The abundance of *Osilobacterium* spp. also increased among females, despite a weaker correlation with Dim 2 ($r=0.48$, $p=0.003$). Gender-based contrasts in dietary patterns were detected in Dim 4 (7.80%) and Dim 5 (6.99%), with females consuming more bread in Dim 4 and more milk in Dim 5 than males. Further PERMANOVA demonstrated that geographic location significantly affected both dietary habits ($F_{by\ margin}=2.13$, $p=0.01$) and gut microbiota ($F_{by\ margin}=11.50$, $p=0.001$) within the Lahu ethnicity, after adjusting for all other factors (gender, BMI, and SBP). Although no significant multivariate dispersion was detected for gut microbiota with respect to geography, unequal variance in dietary habits was observed ($F_{PERMDISP}=13.09$, $p=0.002$). This indicated that dietary behaviors varied greatly among individuals in Chiang Rai Province (particularly within the LahuCR group) (S9 Fig in S1 Text), and that part of the observed differences in dietary patterns could be attributed to this dispersion effect. Additionally, participant characteristics (gender, BMI, and SBP) showed no independent effect on either gut microbiota composition or dietary habits. However, the effect of geographic location on both gut microbiota composition ($F_{interaction}=4.71$, $p=0.001$) and dietary habits ($F_{interaction}=1.70$, $p=0.003$) varied depending on gender across the two locations, indicating a significant interaction effect between geographic location and gender.

When examining the contribution of all variables to individual variations, geographic location was associated with differences in both microbial abundance and dietary habits, with a relatively stronger association with gut microbiota composition. However, substantial dietary variability within the Lahu group should be considered in contextualizing these findings.

### Independent effects of ethnicity and geographical location on dietary habits and gut microbiota

Both ethnicity and geographic location were independently included in the MFA model, along with participant characteristics, dietary habits, and gut microbiota, to assess their independent effects and draw broader conclusions. MFA indicated that both factors influenced variations in dietary habits and gut microbiota; however, ethnicity was more closely associated with gender in explaining dietary variations in a multidimensional space. Differences between individuals from Chiang Rai and Chiang Mai Provinces were mainly attributed to higher gut microbiota abundances in the former and greater grain consumption in the latter, as reflected in Dim 1 (18.79%; Fig 5). Ethnicity contributed to variations in specific taxa and dietary habits, particularly in Dim 2 (8.40%) and Dim 3 (6.89%), respectively. In Dim 3, both gender and ethnicity influenced energy drink consumption, which was higher in the Lahu group irrespective of geographic location. This relation

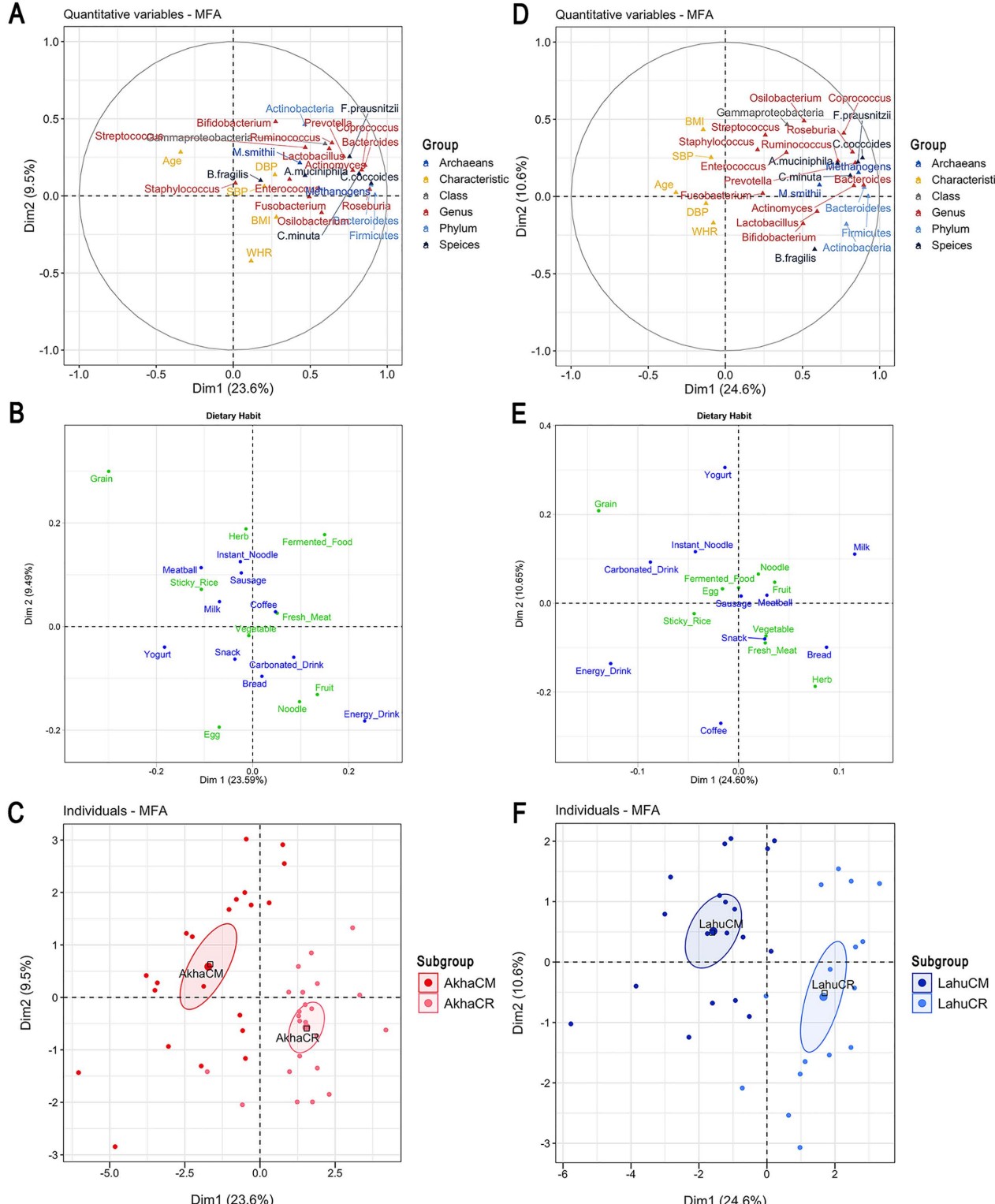

**Fig 4. Multiple factor analysis (MFA) integrating gut microbiota, dietary habits, and participant characteristics between geographic locations.** (A) Correlation plot illustrating the correlation between quantitative variables (microbial taxa and participant characteristics) and dimensions in the Akha ethnic group. (B) Correlation plot illustrating the correlation between dietary habits and dimensions in the Akha ethnic group. (C) Factor map displaying

individual profiles grouped by geographic locations within Akha ethnicity. (D) Correlation plot illustrating the correlation between quantitative variables (microbial taxa and participant characteristics) and dimensions in the Lahu ethnic group. (E) Correlation plot illustrating the correlation between dietary habits and dimensions in the Akha ethnic group. (F) Factor map displaying individual profiles grouped by geographic locations within the Lahu ethnic group. Dim = dimension; Red = AkhaCM; Light red = AkhaCR; Blue = LahuCM; Light blue = LahuCR; CM = Chiang Mai; CR = Chiang Rai.

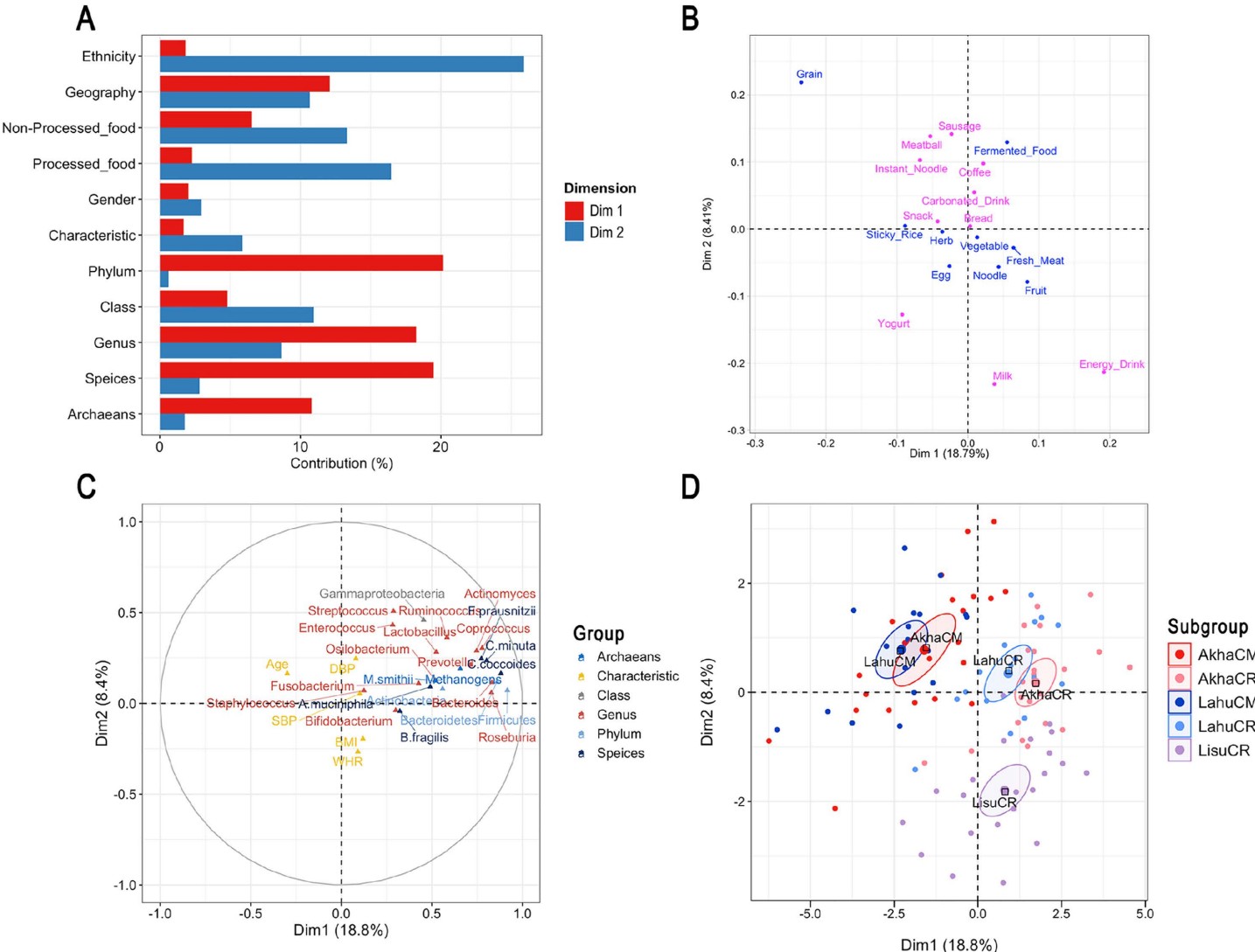

**Fig 5. Multiple factor analysis (MFA) integrating gut microbiota, dietary habits, and participant characteristics across ethnicities and geographic locations.** (A) Bar plot showing the contribution of groups to Dimensions 1 and 2 (Dim 1 and Dim 2). (B) Correlation plot illustrating the relationship between dietary habits and dimensions. (C) Correlation plot illustrating the correlation between quantitative variables (microbial taxa and participant characteristics) and dimensions. (D) Factor map displaying individual profiles grouped by ethnicity and geographic location. Dim = dimension; Red = AkhaCM; Light red = AkhaCR; Blue = LahuCM; Light blue = LahuCR; Light purple = LisuCR; CM = Chiang Mai; CR = Chiang Rai.

became more distinct when considering gender, revealing contrasting consumption patterns between males and females. In Dim 4 (5.82%), an inverse relation in bread consumption was observed between the Lahu and Akha groups. The consumption of this dietary component also exhibited gender-based differences, although gender was less influential than

ethnicity. PERMANOVA confirmed a substantial independent effect of geographic location on gut microbiota ($F_{by\ margin}$ = 12.96, $p$ = 0.001) after adjusting for all other factors (ethnicity, age, gender, BMI, SBP, and DBP), with no significant dispersion. Overall, both geographic location and ethnicity were associated with variations in gut microbiota and dietary habits, with geographic location showing a stronger association with gut microbiota composition.

## Discussion

To the best of our knowledge, this is the first study in Thailand that investigates the influence of both ethnicity and geographic location on gut microbiota composition using absolute quantification (qPCR) of 24 bacterial taxa, while also considering dietary habits and participant characteristics. We collected stool samples from 102 unrelated individuals belonging to Tibeto-Burman-speaking hill-tribe groups residing in Chiang Mai and Chiang Rai Provinces in Northern Thailand. We examined how these factors were associated with gut microbiota profiles across different ethnicities and geographic locations. To minimize confounding effects, we controlled for ethnic backgrounds (Akha, Lahu, and Lisu) and geographic locations (Chiang Mai and Chiang Rai Provinces). Our results revealed that both geographic location and ethnicity were associated with gut microbiota composition, with geographic location exhibiting a relatively stronger association.

Ethnicity has been widely recognized as a significant factor influencing the composition of human gut microbiota [11,12,46]. Our findings support this, revealing that while overall gut microbiota composition in Chiang Mai Province did not differ significantly by ethnicity, specific microbial taxa showed notable differences. In contrast, in Chiang Rai Province, ethnicity had a clear impact on both overall gut microbiota composition and specific taxa. These results align with broader trends identified in larger population studies. For instance, Deschasaux et al. (2018) reported ethnicity-associated differences in gut microbiota among Dutch, Ghanaian, Moroccan, African Surinamese, South-Asian Surinamese, and Turkish populations residing in Amsterdam, the Netherlands [16]. Similarly, Dwiyanto et al. (2021), observed gut microbiota variations among Chinese, Indian, Malay, and Jakun individuals in Southern Peninsular Malaysia [15]. Furthermore, both direct statistical testing and PLS-DA consistently identified Actinobacteria as more prevalent in AkhaCM, and *Actinomyces* (belonging to the Actinobacteria phylum) as notably enriched in the AkhaCR group, while it was least abundant in LisuCR. These findings are consistent with a study comparing gut bacterial communities between Han and Tibetan populations on the Tibetan Plateau, which reported differing abundances of Actinobacteria between these two ethnic groups [47]. Given that our study population was drawn from communities residing at higher altitudes (> 600 m above sea level), these converging findings suggest that the relationship between ethnicity and gut microbiota may be influenced by environmental factors, which could contribute to differentiation in microbial composition at specific taxonomic levels.

Multivariate analyses further indicated that ethnicity was more strongly associated with dietary habits than with gut microbiota composition. For instance, the Akha ethnic group tended to consume higher amounts of fresh meat than the Lahu ethnic groups across both geographic locations, suggesting that these dietary patterns might be influenced by ethnic identity. Furthermore, the consumption of non-processed meat showed contrasting associations with specific microbial taxa in these two ethnic subgroups (positive with *M. smithii* and *F. prausnitzii* in AkhaCR, but negative with *C. minuta* in LahuCM). These results imply that the relationship between gut microbiota and dietary habits may be influenced by ethnic-specific factors (e.g., cultural practice), as seen in previous studies [48], in which different levels of gut bacteria (*Lachnospira* and *Ruminococcus-1*) across racial/ethnic groups were linked with opposing effects of vegetable and red meat consumption on microbial abundance. While such associations can fluctuate across studies, our findings highlight how ethnic-specific dietary behaviors can differentially influence gut microbiota across populations.

Our findings demonstrated that geographic location was associated with gut microbiota composition across both the Akha and Lahu ethnic groups. The greater intra-ethnic variation in gut microbiota across geographic regions, compared to inter-ethnic variation within the same region, aligns with previous studies on Han and seven ethnic minority groups [49], as well as with a large-scale study involving Tibetan, Han, Hui, Miao, Mongolian, Uygur, Bai, Naxi, and Dong groups in China [21]. These studies have consistently reported that geographic location is an important factor associated

with gut microbiota composition, even when controlling for ethnicity. Furthermore, clear microbial differences emerged between participants residing in Chiang Rai and Chiang Mai Provinces for both ethnic groups. Specifically, both AkhaCR and LahuCR had elevated bacterial abundance among the top five taxa (Bacteroidetes, *Streptococcus*, *Coprococcus*, *Prevotella*, and *Bacteroides*) when compared to their respective counterparts in Chiang Mai Province. Although the Akha and Lahu have distinct cultural backgrounds, this shared pattern may reflect common environmental exposures (e.g., living at elevations > 900 m above sea level) in Chiang Rai Province. However, dietary practices and food availability were not assessed in this study. As Lin et al. (2020) suggested that regional or ethnic characteristics tend to become more similar over time due to the mutual adoption of local dietary cultures, further investigation is needed to clarify the potential role of these factors [21].

Beyond geographic influences alone, the integration of dietary data further revealed that diet-geography interactions were associated with differences in gut microbiota. In the Akha ethnic group, we identified a negative association between grain consumption and gut microbial abundance, with lower grain intake linked to higher microbial abundance in AkhaCR, while the opposite trend was observed for AkhaCM. Our results are consistent with those of prior studies showing that geographic relocation–such as migration from Thailand to the US–can induce notable gut microbiome shifts due to changes in dietary patterns, beyond the influence of ethnicity alone [50]. Although LahuCR exhibited greater variability in dietary behaviors compared to LahuCM, which could potentially mask the effect of geographic location, this intra-group variation did not influence the gut microbiota. The greater dietary variability observed in LahuCR may be attributed to lifestyle differences; however, the effect of geographic location on gut microbiota remained unaffected. This indicated that the observed association between geographic location and gut microbiota composition may persist despite variations in diet; however further investigation is needed to clarify this relationship. Furthermore, in the Akha ethnic group, both age and gender interacted with geographic location to explain part of the variability in microbiota and diet. In contrast, among the Lahu, only gender interacted with geographic location. This suggested that demographic factors may modulate the relationship between geography and gut microbiota and/or dietary habits in an ethnic-specific manner. Collectively, our study indicates that geographic location was more prominently associated with variations in gut microbiota, with dietary habits acting as key mediators, particularly within the Akha ethnic group.

Finally, our study demonstrated that variations in gut microbiota were primarily associated with geographic differences, whereas ethnicity was more closely linked to dietary patterns. Higher gut microbiota abundance in individuals from Chiang Rai Province, suggested that environmental factors tied to specific regions may influence microbial abundance. These results contribute to a growing body of literature emphasizing that regional factors, including environmental exposures and dietary consumption patterns, play a dominant role in shaping gut microbial communities, often surpassing the effects of ethnic identity [46,49]. Participant characteristics, particularly gender, emerged as relevant factors that interacted with ethnicity and thus contributed to variations in dietary patterns. However, in comparison to the prominent effect of geographic location on gut microbiota, the influence of participant characteristics was relatively limited.

A key strength of this study is its design, which enabled us to disentangle the independent effects of ethnicity and geographic location on gut microbiota composition in populations residing in rural mountainous areas of Northern Thailand. By recruiting participants who self-identify with the same ethnic background for at least three generations, we were able to control for host genetic and cultural backgrounds, thereby reducing confounding influences and improving the robustness of our findings. However, several limitations should be acknowledged. First, the relatively small sample size, which resulted from the reliance on voluntary participation within a specific population (the Tibeto-Burman hill tribes), may have limited the statistical power to detect subtle effects. In addition, the age range of participants was random, introducing potential age-related variation that could influence both dietary habits and gut microbiota. Second, while dietary data were collected based on consumption frequency, detailed information on specific food items, portion sizes, and nutrient content was lacking, which may have limited our ability to fully capture the relationship between gut microbiota and dietary behaviors. Third, environmental exposures such as water sources [51], sanitation [52], and occupational activities [53], which

could influence gut microbiota composition and potentially confound the effects of geography should be incorporated into future studies. Finally, although qPCR enabled sensitive and specific quantification of targeted bacterial taxa [54], this approach limited our ability to capture the full breadth of microbial diversity and community composition. Therefore, this study provides a preliminary and focused overview of key gut microbes in relation to ethnicity and geography. Future studies should incorporate next-generation sequencing approaches to achieve a more comprehensive understanding of gut microbial diversity and function.

To deepen our understanding of the observed patterns, future research with larger sample sizes should incorporate additional host-related factors, such as genetic background, blood chemistry, lifestyle patterns, socio-economic status, and cultural practices all of which may be related to the gut microbiota in these populations.

## Conclusion

We employed an absolute quantification (qPCR) approach to examine the abundance of gut microbiota among Tibeto-Burman-speaking hill-tribe populations in Northern Thailand and explored how ethnicity and geographic location may influence gut microbiota composition. Both direct comparisons and multivariate analyses indicated that geographic location was the most strongly associated with variations in gut microbiota, while ethnicity appeared to be more closely linked to differences in dietary habits and, to a lesser extent, gut microbiota composition across the two investigated regions (Chiang Mai and Chiang Rai Provinces). Gut microbiota profiles tended to be more similar between ethnic groups within the same location than among individuals of the same ethnicity residing in different regions. Considering dietary behaviors and participant characteristics, dietary habits varied in their association with gut microbiota across ethnic and geographic groups. Host factors such as age, gender, and BMI had a comparatively smaller influence on microbial variation in this study.

## Supporting information

**S1 Text. S1 Fig.** Partial least squares discriminant analysis (PLS-DA) of gut microbiota across ethnic groups. Discriminant analysis identifying key features (microbial taxa) associated with each ethnic group in Chiang Mai province, where the median values are maximized along Component 1 (A) and Component 2 (B). (C) Discriminant analysis identifying key features (microbial taxa) associated with each ethnic group in Chiang Rai province, where the median values are maximized along Component 1. Bar lengths represent the loading weights, reflecting the contribution of each taxon (ranked from bottom to top). Red=AkhaCM; Light red=AkhaCR; Blue=LahuCM; Light blue=LahuCR; Light purple=LisuCR; CM=Chiang Mai; CR=Chiang Rai. **S2 Fig.** Multiple factor analysis (MFA) integrating gut microbiota and dietary habits between ethnic groups in Chiang Mai Province. (A) Bar plot showing the contribution of groups to Dimensions 1 and 2 (Dim 1 and Dim 2). (B) Correlation plot illustrating the relationship between dietary habits and dimensions. (C) Correlation circle plot showing the association between quantitative variables (microbial taxa) and dimensions. (D) Factor map displaying individual profiles grouped by ethnicity. Red=AkhaCM; Blue=LahuCM; CM=Chiang Mai; Dim=dimension. **S3 Fig.** Multiple factor analysis (MFA) integrating gut microbiota and dietary habits across ethnic groups in Chiang Rai Province. (A) Bar plot showing the contribution of groups to Dimensions 1 and 2 (Dim 1 and Dim 2). (B) Correlation plot illustrating the relationship between dietary habits and dimensions. (C) Correlation circle plot showing the association between quantitative variables (microbial taxa) and dimensions. (D) Factor map displaying individual profiles grouped by ethnicity. Light red=AkhaCR; Light blue=LahuCR; Light purple=LisuCR; CR=Chiang Rai; Dim=dimension. **S4 Fig.** Multiple factor analysis (MFA) integrating gut microbiota and dietary habits, and participant characteristics between ethnic groups in Chiang Mai Province. (A) Bar plot showing the contribution of groups to Dimensions 1 and 2 (Dim 1 and Dim 2). (B) Correlation plot illustrating the relationship between dietary habits and dimensions. (C) Correlation circle plot showing the association between quantitative variables (microbial taxa and participant characteristics) and dimensions. (D) Factor map displaying individual profiles grouped by ethnicity. Red=AkhaCM; Blue=LahuCM; CM=Chiang Mai; Dim=dimension. **S5 Fig.** Beta dispersion analysis

based on a Euclidean distance matrix illustrating variation in the dispersion of dietary behaviors between two ethnic groups in Chiang Mai province. A: ordination plot, B: PCoA coordinate plot. Red = AkhaCM; Blue = LahuCM; CM = Chiang Mai. **S6 Fig.** Multiple factor analysis (MFA) integrating gut microbiota and dietary habits, and participant characteristics between ethnic groups in Chiang Rai Province. (A) Bar plot showing the contribution of groups to Dimensions 1 and 2 (Dim 1 and Dim 2). (B) Correlation plot illustrating the relationship between dietary habits and dimensions. (C) Correlation circle plot showing the association between quantitative variables (microbial taxa and participant characteristics) and dimensions. (D) Factor map displaying individual profiles grouped by ethnicity. Light red = AkhaCR; Light blue = LahuCR; Light purple = LisuCR; CR = Chiang Rai; Dim = dimension. **S7 Fig.** Multiple factor analysis (MFA) integrating gut microbiota and dietary habits between geographic locations within the Akha ethnic group. (A) Bar plot showing the contribution of groups to Dimensions 1 and 2 (Dim 1 and Dim 2). (B) Correlation plot illustrating the relationship between dietary habits and dimensions. (C) Correlation circle plot showing the association between quantitative variables (microbial taxa) and dimensions. (D) Factor map displaying individual profiles grouped by geographic location. Red = AkhaCM; Light red = AkhaCR; CM = Chiang Mai; CR = Chiang Rai; Dim = dimension. **S8 Fig.** Multiple factor analysis (MFA) integrating gut microbiota and dietary habits between geographic locations within the Lahu ethnic group. (A) Bar plot showing the contribution of groups to Dimensions 1 and 2 (Dim 1 and Dim 2). (B) Correlation plot illustrating the relationship between dietary habits and dimensions. (C) Correlation circle plot showing the association between quantitative variables (microbial taxa) and dimensions. (D) Factor map displaying individual profiles grouped by geographic location. Blue = LahuCM; Light blue = LahuCR; CM = Chiang Mai; CR = Chiang Rai; Dim = dimension. **S9 Fig.** Beta dispersion analysis based on a Euclidean distance matrix illustrating the variation in dispersion of dietary behaviors between the LahuCM and LahuCR groups. A: ordination plot, B: PCoA coordinate plot. Blue = LahuCM; Light blue = LahuCR; CM = Chiang Mai; CR = Chiang Rai.
(ZIP)

**S2 Text.** **S1 Table.** Primer pairs targeting bacterial 16S rRNA genes. **S2 Table.** Mean differences in gut microbiota abundance between ethnic groups within Chiang Mai Province. **S3 Table.** Mean differences in gut microbiota abundance between ethnic groups within Chiang Rai Province. **S4 Table.** Median differences in dietary consumption frequencies between ethnic groups within Chiang Mai Province. **S5 Table.** Median differences in dietary consumption frequencies between ethnic groups within Chiang Rai Province. **S6 Table.** Significant Spearman's rank Correlations ($p < 0.05$) between gut microbiota and dietary habits in Chiang Mai province. **S7 Table.** Significant Spearman's rank Correlations ($p < 0.05$) between gut microbiota and dietary habits in Chiang Rai province. **S8 Table.** Mean differences in gut microbiota abundance between geographic locations within the Akha ethnic group. **S9 Table.** Mean differences in gut microbiota abundance between geographic locations within the Lahu ethnic group. **S10 Table.** Median differences in dietary consumption frequencies between geographic locations within the Akha ethnic group. **S11 Table.** Median differences in dietary consumption frequencies between geographic locations within the Lahu ethnic group.
(ZIP)

**S3 Text.** **S1 File.** Raw data for downstream analyses. **S2 File.** Raw data and MFA results on associations between gut microbiota and dietary habits across ethnicities and geographic locations. **S3 File.** Raw data and MFA results on associations between gut microbiota, dietary habits, and participant characteristics across ethnicities and geographic locations. **S4 File.** Raw data and PERMANOVA of gut microbiota and dietary habits across ethnicities and geographic locations.
(ZIP)

## Acknowledgments

We would like to thank all volunteers for providing fecal samples, anthropometric measurements, and dietary behavior data from five villages in Chiang Mai and Chiang Rai provinces. Special thanks to Mr. Suwapat Satupak, Mrs. Maneesawan Dansawan, and Mr. Chayapa Sombat for their technical assistance in collecting samples.

## Author contributions

**Conceptualization:** Angkhana Inta, Metawee Srikummool, Jatupol Kampuansai, Siam Popluechai.

**Data curation:** Tanapon Seetaraso.

**Formal analysis:** Tanapon Seetaraso, Lucsame Gruneck, Phatthanaphong Therdtatha, Siam Popluechai.

**Funding acquisition:** Angkhana Inta.

**Investigation:** Tanapon Seetaraso, Vasana Jinatham, Sitanan Kantakat, Marie-Lou Albert, Phatthanaphong Therdtatha, Angkhana Inta, Metawee Srikummool, Jatupol Kampuansai.

**Methodology:** Jatupol Kampuansai, Siam Popluechai.

**Project administration:** Siam Popluechai.

**Resources:** Siam Popluechai.

**Supervision:** Jatupol Kampuansai, Siam Popluechai.

**Validation:** Jatupol Kampuansai, Siam Popluechai.

**Visualization:** Tanapon Seetaraso, Lucsame Gruneck.

**Writing – original draft:** Tanapon Seetaraso, Lucsame Gruneck, Phatthanaphong Therdtatha, Jatupol Kampuansai, Siam Popluechai.

**Writing – review & editing:** Tanapon Seetaraso, Lucsame Gruneck, Phatthanaphong Therdtatha, Angkhana Inta, Metawee Srikummool, Jatupol Kampuansai, Siam Popluechai.

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
