## [Decision Letter · Decision Letter 0]

3 Jul 2025

PONE-D-25-17528Effects of ethnicity and geography on the fecal microbiota and dietary habits of Tibeto-Burman hill tribes in Northern ThailandPLOS ONE

Dear Dr. Popluechai,

Thank you for submitting your manuscript to PLOS ONE. After careful consideration, we feel that it has merit but does not fully meet PLOS ONE’s publication criteria as it currently stands. Therefore, we invite you to submit a revised version of the manuscript that addresses the points raised during the review process.

We look forward to receiving your revised manuscript.

Kind regards,

KAMARAJ RAJU, PhD

Academic Editor

PLOS ONE

Journal Requirements:

4. We note that Figure 1 in your submission contain map/satellite images which may be copyrighted. All PLOS content is published under the Creative Commons Attribution License (CC BY 4.0), which means that the manuscript, images, and Supporting Information files will be freely available online, and any third party is permitted to access, download, copy, distribute, and use these materials in any way, even commercially, with proper attribution. For these reasons, we cannot publish previously copyrighted maps or satellite images created using proprietary data, such as Google software (Google Maps, Street View, and Earth). For more information, see our copyright guidelines: http://journals.plos.org/plosone/s/licenses-and-copyright.

Additional Editor Comments:

Dear Dr. Siam Popluechai,

Thank you for submitting your manuscript entitled "Effects of ethnicity and geography on the fecal microbiota and dietary habits of TibetoBurman hill tribes in Northern Thailand" to Plos One.

We have now received reports from two reviewers:

• Reviewer 1 has recommended a minor revision, noting a few areas where clarification or small adjustments would enhance the manuscript.

• Reviewer 2 has recommended the manuscript for acceptance in its current form, indicating satisfaction with the overall quality and contribution of your work.

Based on these reviews, the editorial team has decided to request a minor revision before we proceed with final acceptance. We kindly ask you to address the comments raised by Reviewer 1 and submit a revised version of your manuscript. Please include a detailed point-by-point response to the reviewer’s suggestions, indicating the changes made or explaining your rationale if no changes were made.

Once the revised manuscript is received, we will proceed to the next step, which may include a brief re-evaluation to confirm that the concerns have been adequately addressed.

We appreciate your contribution to the journal and look forward to receiving your revised manuscript soon.

Best regards,

Kamaraj Raju

Academic Editor

Plos One

Reviewers' comments:

Reviewer's Responses to Questions

**Comments to the Author**

1. Is the manuscript technically sound, and do the data support the conclusions?

Reviewer #1: Yes

Reviewer #2: Yes

2. Has the statistical analysis been performed appropriately and rigorously? 

Reviewer #1: Yes

Reviewer #2: Yes

3. Have the authors made all data underlying the findings in their manuscript fully available?

Reviewer #1: Yes

Reviewer #2: Yes

4. Is the manuscript presented in an intelligible fashion and written in standard English?

Reviewer #1: Yes

Reviewer #2: Yes

5. Review Comments to the Author

Reviewer #1: Reviewer Comments

This manuscript presents a comprehensive and well-performed study investigating influences of ethnicity and geography on gut microbiota composition and dietary intake with respect to Tibeto-Burman hill-tribe populations in Northern Thailand. Quantitative PCR, multivariate statistics (MFA, PLS-DA), and the sampling from carefully defined ethnic and geographic subgroups lend weight to this field. The manuscript is straightforward, the figures informative, and the statistical approaches proper. However, there are issues that should be addressed to strengthen the manuscript prior to publication.

Comments

- While qPCR could be considered an exact technique-matters for numbers, the authors would be wise to elucidate why they opted for qPCR instead of the more common 16S rRNA sequencing, especially concerning taxonomic range and resolution. Were there any drawbacks with taxonomic profiling?

- A total of 102 subjects divided into various subgroups (3 ethnic groups × 2 locations) renders per-group samples quite small for statistically conclusive interpretations. Please consider conducting a power analysis or discussing the limitations presented by such small group sizes.

- The manuscript tends to assert causal relationships between geography and gut microbiota composition. However, strong causal claims are not supportable due to the cross-sectional design of the study. It is requested that the language state association rather than causality.

- Environmental exposures beyond what has been considered in diet and elevation have not been discussed (i.e., water sources, sanitation, occupational activities). These exposures could alter the gut microbiota and potentially confound effects of geography. Consider mentioning this as a limitation.

- The authors state that all relevant data are available in the manuscript and supporting information. The authors should also please consider making raw qPCR data and all metadata used in the multivariate analyses

- Clarify how the participants were confirmed to have no kinship relation and to be representative of their ethnic groups.

- The manuscript also states that illiterate participants signed a consent form with a fingerprint. Please specify if this procedure is aligned with national and international procedures for ethical consideration.

- Heatmaps and similar figures (e.g., Figs 2 and 3) require better contrast and simpler and clearer legends to be more interpretable.

- Use the same ethnic group names consistently throughout (e.g., "AkhaCM" vs. "Akha CM") and explain the abbreviations at first use both in the text and in figure legends.

- Some sentences could be clearer (e.g., lines 295–298). A thorough proofreading is recommended.

Recommendation: Minor Revision

This study addresses an important gap in the literature and provides valuable insights into how geography and ethnicity influence gut microbiota in underrepresented populations. With minor revisions and clarifications, the manuscript would be suitable for publication.

Reviewer #2: Dear Editor,

I have carefully reviewed the manuscript titled "Effects of ethnicity and geography on the fecal microbiota and dietary habits of Tibeto-Burman hill tribes in Northern Thailand" submitted to Plos One. The authors have presented a well structured and scientifically sound study that contributes valuable insights to the field of Gut Microbiome Study.

Strengths of the Manuscript:

• The research question is clearly defined and relevant.

• The methodology is appropriate and well executed.

• Data analysis is thorough and supports the conclusions drawn.

• The discussion effectively contextualizes the findings with existing literature.

• The manuscript is well written and easy to follow.

Recommendation: I recommend acceptance of this manuscript in its current form / with minor editorial revisions (grammar, formatting), if needed by the journal.

Congratulations to the authors on their work.

Sincerely,

Sivagami Subramanian

6. PLOS authors have the option to publish the peer review history of their article (what does this mean? ). If published, this will include your full peer review and any attached files.

**Do you want your identity to be public for this peer review?** For information about this choice, including consent withdrawal, please see our Privacy Policy .

Reviewer #1: **Yes: ** Pratheep Thangaraj

Reviewer #2: No

---

## [Author Response · Author response to Decision Letter 1]

30 Jul 2025

Editor

Response: We have carefully reviewed the manuscript and supporting files to ensure that all formatting and file naming conventions comply with PLOS ONE’s style requirements, as outlined in the journal’s guidelines.

Response: We have removed all funding-related information from the manuscript and have provided the relevant details exclusively in the Funding Statement section of the online submission form, in accordance with the journal’s requirements.

Response: We appreciate the reminder. We have successfully verified the corresponding author’s ORCID iD and updated our information in Editorial Manager as instructed.

4. We note that Figure 1 in your submission contain map/satellite images which may be copyrighted. All PLOS content is published under the Creative Commons Attribution License (CC BY 4.0), which means that the manuscript, images, and Supporting Information files will be freely available online, and any third party is permitted to access, download, copy, distribute, and use these materials in any way, even commercially, with proper attribution. For these reasons, we cannot publish previously copyrighted maps or satellite images created using proprietary data, such as Google software (Google Maps, Street View, and Earth).

Response: We were unable to obtain permission from the original copyright holder to publish the map/satellite images in Figure 1 under the CC BY 4.0 license. Therefore, we have removed Figure 1 and replaced it with Table 1, which presents the relevant sampling information.

Response: We have carefully reviewed our reference list. All references remain current, relevant, and valid for the scientific context of our study.

Reviewer 1

This manuscript presents a comprehensive and well-performed study investigating influences of ethnicity and geography on gut microbiota composition and dietary intake with respect to Tibeto-Burman hill-tribe populations in Northern Thailand. Quantitative PCR, multivariate statistics (MFA, PLS-DA), and the sampling from carefully defined ethnic and geographic subgroups lend weight to this field. The manuscript is straightforward, the figures informative, and the statistical approaches proper. However, there are issues that should be addressed to strengthen the manuscript prior to publication.

Response: We are grateful for the reviewer’s thoughtful feedback and careful evaluation of our manuscript. We sincerely thank you for the insightful and constructive comments and suggestions, which have significantly helped us improve the overall structure and clarity of the manuscript. Below, we provide a point-by-point response to each comment.

Comments

1. While qPCR could be considered an exact technique-matters for numbers, the authors would be wise to elucidate why they opted for qPCR instead of the more common 16S rRNA sequencing, especially concerning taxonomic range and resolution. Were there any drawbacks with taxonomic profiling?

Response: We thank the reviewer for this insightful comment. As this study aimed to quantify and compare specific gut bacteria of interest across different ethnic groups and geographic locations, we employed a qPCR-based approach, which enabled rapid detection with high sensitivity and specificity (Yan et al., 2024). This method provided a preliminary overview of how ethnic and geographic factors may influence gut microbiota in these underrepresented communities. While qPCR provides high sensitivity and specificity for selected taxa, we acknowledge that qPCR does not provide a comprehensive overview of the microbial community structure or diversity, as would be achievable through 16S rRNA gene sequencing, which offers broader taxonomic resolution. We have discussed this as a limitation of our study in the Discussion section (lines 661-666).

2. A total of 102 subjects divided into various subgroups (3 ethnic groups × 2 locations) renders per-group samples quite small for statistically conclusive interpretations. Please consider conducting a power analysis or discussing the limitations presented by such small group sizes.

Response: We appreciate the reviewer’s comment regarding the small sample size within each subgroup and its implications for statistical interpretation. We conducted a power analysis for group comparisons using the pwr package in R. The analysis indicated that approximately 64 participants per group (for two-group comparisons) and 52 participants per group (for three-group comparisons) would be required to detect a medium effect size with 80% power at a significance level of 0.05. However, recruitment for this study was based on voluntary participation from specific communities, namely the Tibeto-Burman hill tribes, which resulted in a smaller sample size than the calculated requirement. This limitation, which stems from reliance on voluntary participation within a specific population, has been acknowledged in the revised manuscript (lines 651-654).

3. The manuscript tends to assert causal relationships between geography and gut microbiota composition. However, strong causal claims are not supportable due to the cross-sectional design of the study. It is requested that the language state association rather than causality.

Response: We greatly appreciate the reviewer’s thoughtful comment. We agree that, given the cross-sectional design of our study, the language should reflect associations rather than causal relationships. In response to this valuable suggestion, we have carefully revised the language throughout the manuscript to ensure that statements regarding geography and gut microbiota composition are presented as associations rather than implying causality.

4. Environmental exposures beyond what has been considered in diet and elevation have not been discussed (i.e., water sources, sanitation, occupational activities). These exposures could alter the gut microbiota and potentially confound effects of geography. Consider mentioning this as a limitation.

Response: Thank you for raising this important point. We agree that environmental exposures beyond diet and elevation, such as water sources, sanitation, and occupational activities, may serve as potential confounders influencing gut microbiota composition. We have acknowledged this as a limitation of our study and highlighted the importance of incorporating these variables in future research (lines 658-661).

5. The authors state that all relevant data are available in the manuscript and supporting information. The authors should also please consider making raw qPCR data and all metadata used in the multivariate analyses

Response: Thank you for your valuable suggestion. We have provided raw qPCR data and all relevant metadata used in the multivariate analyses in S1-S4 Files.

6. Clarify how the participants were confirmed to have no kinship relation and to be representative of their ethnic groups.

Response: Thank you for raising this important point. We confirmed that participants had no close kinship relationships and were representative of their respective ethnic groups through a sampling process aligned with a previous study (Kampuansai et al., 2023). The selection of ethnic groups was guided by historical and ethnographic data, including detailed accounts of cultural identity and community organization from Inta et al., 2023, as well as databases from the Health Center for Ethnic Group, Marginal People and Migrant Worker (HHDC), and the Princess Maha Chakri Sirindhorn Anthropology Centre (SAC). Participant recruitment involved direct interviews with volunteers and consultations with local community leaders who were knowledgeable about family lineages and community structure. This process ensured that participants were not closely related to others in the study for at least three generations. We have clarified this procedure in the revised Materials and Methods section (lines 112-116).

7. The manuscript also states that illiterate participants signed a consent form with a fingerprint. Please specify if this procedure is aligned with national and international procedures for ethical consideration.

Response: Thank you for highlighting this important point. We confirm that the procedure for obtaining informed consent from illiterate participants using fingerprint signatures strictly adhered to all relevant ethical guidelines and regulations, in accordance with the Declaration of Helsinki. The consent form was verbally explained to each participant in their native language, and an impartial witness was present throughout the process. The witness also signed a statement confirming that the participant understood the information and voluntarily agreed to participate. We have expanded the Materials and Methods section to clarify this procedure (lines 100-105) and have uploaded the translated consent form for reference.

8. Heatmaps and similar figures (e.g., Figs 2 and 3) require better contrast and simpler and clearer legends to be more interpretable.

Response: We have updated the figure numbers, which are now presented as Figure 1–2. In addition, we have simplified and clarified the figure legends to improve their interpretability for readers.

9. Use the same ethnic group names consistently throughout (e.g., "AkhaCM" vs. "Akha CM") and explain the abbreviations at first use both in the text and in figure legends.

Response: The ethnic group names in the revised manuscript have been corrected for consistency throughout the text, and all abbreviations have been clearly explained at their first occurrence in both the main text and figure legends.

10. Some sentences could be clearer (e.g., lines 295–298). A thorough proofreading is recommended.

Response: We thank the reviewer for the suggestion. The sentences have been revised to improve clarity and readability, as recommended (lines 316-319). Additionally, the manuscript has undergone professional proofreading by Editage. An editing certificate has been included for your reference.

Reviewer 2

I have carefully reviewed the manuscript titled "Effects of ethnicity and geography on the fecal microbiota and dietary habits of Tibeto-Burman hill tribes in Northern Thailand" submitted to Plos One. The authors have presented a well-structured and scientifically sound study that contributes valuable insights to the field of Gut Microbiome Study.

Strengths of the Manuscript:

• The research question is clearly defined and relevant.

• The methodology is appropriate and well executed.

• Data analysis is thorough and supports the conclusions drawn.

• The discussion effectively contextualizes the findings with existing literature.

• The manuscript is well written and easy to follow.

Recommendation: I recommend acceptance of this manuscript in its current form / with minor editorial revisions (grammar, formatting), if needed by the journal.

Congratulations to the authors on their work.

Response: We greatly appreciate the reviewer for your thorough evaluation and kind feedback. We are grateful for the positive comments regarding the clarity of our research question, the appropriateness of our methodology, and the strength of our analysis and discussion. To ensure clarity and consistency, the manuscript has been professionally proofread by Editage A certificate of editing has been included as an attachment for your reference. Additionally, the formatting has been revised to ensure consistency, as previously mentioned by the editor.

---

## [Editor Report · Decision Letter 1]

27 Aug 2025

Effects of ethnicity and geography on the fecal microbiota and dietary habits of Tibeto-Burman hill tribes in Northern Thailand

PONE-D-25-17528R1

Dear Dr. Popluechai,

We’re pleased to inform you that your manuscript has been judged scientifically suitable for publication and will be formally accepted for publication once it meets all outstanding technical requirements.

Kind regards,

KAMARAJ RAJU, PhD

Academic Editor

PLOS ONE

Additional Editor Comments (optional):

Dear Siam Popluechai,

As the Academic Editor responsible for handling your submission, I am pleased to inform you that your manuscript entitled “Effects of ethnicity and geography on the fecal microbiota and dietary habits of Tibeto-Burman hill tribes in Northern Thailand” has been accepted for publication in PlosOne.

Your work provides significant insights into the role of ethnicity and geography in shaping gut microbiota and dietary practices, particularly among Tibeto-Burman hill tribes in Northern Thailand. The reviewers and I agree that your study will be of considerable interest to readers and makes an important contribution to the field.

Thank you for choosing PlosOne the venue for your research.

With best regards,

Kamaraj Raju

Academic Editor

PlosOne
---

## [Editor Report · Acceptance letter]

PONE-D-25-17528R1

PLOS ONE

Dear Dr. Popluechai,

I'm pleased to inform you that your manuscript has been deemed suitable for publication in PLOS ONE. Congratulations! Your manuscript is now being handed over to our production team.

Kind regards,

on behalf of

Dr. KAMARAJ RAJU

Academic Editor

PLOS ONE